# Acceptance and Preferences of Using Ambient Sensor-Based Lifelogging Technologies in Home Environments

**DOI:** 10.3390/s21248297

**Published:** 2021-12-11

**Authors:** Julia Offermann, Wiktoria Wilkowska, Angelica Poli, Susanna Spinsante, Martina Ziefle

**Affiliations:** 1Human-Computer Interaction Center, RWTH Aachen University, Campus-Boulevard 57, 52074 Aachen, Germany; wilkowska@comm.rwth-aachen.de (W.W.); ziefle@comm.rwth-aachen.de (M.Z.); 2Department of Information Engineering, Università Politecnica delle Marche, 60131 Ancona, Italy; a.poli@staff.univpm.it (A.P.); s.spinsante@staff.univpm.it (S.S.)

**Keywords:** technology acceptance, ambient sensor-based lifelogging technologies, user preferences, different sensor types, assumed costs, willingness to pay

## Abstract

Diverse sensor-based technologies can be used to track (older and frail) people’s movements and behaviors in order to detect anomalies and emergencies. Using several ambient sensors and integrating them into an assisting ambient system allows for the early identification of emergency situations and health-related changes. Typical examples are passive infrared sensors (PIR), humidity and temperature sensors (H&T) as well as magnetic sensors (MAG). So far, it is not known whether and to what extent these three specific sensor types are perceived and accepted differently by future users. Therefore, the present study analyzed the perception of benefits and barriers as well as acceptance of these specific sensor-based technologies using an online survey (reaching N=312 German participants). The results show technology-related differences, especially regarding the perception of benefits. Furthermore, the participants estimated the costs of these sensors to be higher than they are, but at the same time showed a relatively high willingness to pay for the implementation of sensor-based technologies in their home environment. The results enable the derivation of guidelines for both the technical development and the communication and information of assisting sensor-based technologies and systems.

## 1. Introduction

In light of demographic change and an increasingly aging population, assisting lifelogging technologies are developed in order to support people in being as independent and active as possible [1,2]. In fact, lifelogging can help to raise awareness about one’s own quality of life, thanks to the use of wearable trackers, ambient sensors, and even manual entry to collect suitable indicators [3]. The spectrum of the technological developments is extremely broad, reaching from video-based approaches [1,4,5], including those exploiting multiple cameras and suitable artificial intelligence-based processing aimed at fall detection, such as [6,7,8], over ambient systems based on different sensors and technologies (such as microphones, pressure or vibration sensors) [9,10] to wearable technologies using acceleration and rotation rate sensors [11,12,13]. All these technologies and systems have in common that they can be used to support people in their everyday life, e.g., by detecting emergency situations, by identifying typical movement and behavior patterns as well as anomalies, or by remembering functions [2,14]. Beyond complex technical solutions, the usage of ambient room sensors represents a comparatively inexpensive and promising alternative.

In 2018, the review by Uddin et al. [15] analyzed several publications related to projects about the use of ambient sensors for elderly care and independent living. Overall, the technology appeared promising, especially in long-term scenarios, with different types or combinations of sensors used, such as PIR, simple ambient sensors, and radar-based ones. Technical limitations and open issues were identified in terms of durability, communication, and power requirements of the installed sensors. At the same time, the authors found that the assumption of sensor acceptance by the elderly had not been well investigated.

Passive technologies, i.e., sensors and sensing systems that are not based on the use of wearable devices and do not require periodic (e.g., daily) recharge nor interaction or maintenance by the users, such as most of the ambient sensor-based solutions, can be perceived as less intrusive (unless video cameras are included in the system) and may provide more consistent and robust information in the long-term [16]. Their adoption is increasing, not only in relation to monitoring applications targeting elderly or frail subjects, but also in scenarios addressing the well-being and health status of young subjects, such as students in campuses [17]. In many cases, the use of sensors, either wearable or ambient-based ones, is associated with Physical Activity Recognition and Monitoring (PARM) [18], and it may also encompass on-object sensors (exploiting Radio Frequency Identification—RFID—tags), in order to enable the recognition and classification of complex activities requiring interaction with the objects located in the living environment.

Further and besides the technical opportunities and functions, it is currently not known whether and to what extent future users distinguish between different types of sensor-based lifelogging technologies. In addition, it is an open question how high future users estimate the costs for the acquisition and what costs they are willing to pay. Therefore, this paper focuses on the future users’ perspectives on different ambient sensor-based lifelogging technologies. Using an online survey (N=312) the participants’ perception and acceptance of three different sensor types, as well as the assumed costs and the willingness to pay, are investigated.

This paper is structured as follows: In Section 1, a technical overview is given focusing on different ambient sensor types and their functional characteristics. Subsequently, the current research state on the acceptance of (sensor-based) lifelogging technologies is presented. Section 4 includes a description of the empirical approach describing the design of the applied online survey, the sample’s characteristics, and the data analysis. Further, the results of the study are presented (Section 5) and discussed including a reflection of limitations and opportunities for future work (Section 6).

## 2. Sensor-Based Lifelogging Technologies

Within this section, the technical background of the empirical study is presented with a technical overview of sensor-based technologies like PIR, H&T, and MAG sensors as examples.

### 2.1. PIR

While recent and innovative research activities are testing the use of PIR sensors to measure physiological parameters such as heart rate (HR) or breathing rate (BR), as discussed in a recent review by Cetin et al. [19], the most common use of PIRs is in occupancy detection within indoor environments, based on which different automation and control decisions may be taken [20]. Thanks to their relative simplicity and high energy-savings potential (for example when used to control light switch on or off, based on room occupancy detection), PIR sensors are nowadays commonly integrated into new buildings and constructions, but they can also be easily applied in retrofit projects. People emit heat and their motion stimulates the sensor reaction. Motion can be detected by PIRs only in a line-of-sight condition, so the target must be located inside the coverage area of the sensor and directly visible from it. PIRs are typically built with a multi-faceted lens, the characteristics of which determine either the coverage area or the best size of motion the sensor can detect. Usually, PIRs are more sensitive to motion occurring in a lateral position with respect to the sensor, and their sensitivity decreases with increasing distance of the target from the sensor. In our study, a quite small battery-powered PIR sensor was located inside the bathroom, on the top of the mirror, in order to identify the events corresponding to the user entering and exiting the bathroom. According to the specifications provided by the manufacturer, the detectable distance amounts to 7 m, with a 170° field of view. The specific device chosen can be easily found in the market; specifically, it was bought from an online seller, in a complete kit of ambient sensors including the PIR, a few magnetic sensors, a temperature and humidity sensor, and a node acting as a gateway, for collecting the data generated by the sensors on Bluetooth Low Energy (BLE) links, and sending them to a remote online platform usually by a WiFi connection, in a similar fashion as presented in [21].

### 2.2. H&T

An indoor H&T sensor is capable of measuring relative humidity and ambient temperature in indoor environments. The specific model chosen for our study can detect a 0.3 °C temperature variation in the range [0 ÷ 60] °C, and a 3% variation of the relative humidity in the range [0 ÷ 99] %. The sensor has a small display that allows reading the values of the measured quantities directly, and, at the same time, it saves the collected data on the remote server, by sending it to the gateway. H&T sensors are very common and usually adopted in home and building automation systems, to ensure a quite stable indoor comfort condition. In our study, the H&T variations detected by the sensor, based on its location inside the home environment, may allow us to detect specific events like those corresponding to food preparation during the day.

### 2.3. MAG

Magnetic sensors are typically used to detect doors and windows opening and closing. In fact, a magnetic sensor acts as a switch, and it is composed of two parts: the true sensing element is usually placed on or inside the door or window frame; the corresponding magnet is placed on or inside the door or window itself. By opening the door or window, the magnet gets separated from the sensor and activates it. The MAG sensors used in our study are battery-powered and the manufacturer states a battery life of around two years. The same magnetic sensors have been applied to drawers and furniture doors to collect data able to describe the way people interact with the living environment. For example, if the patient usually keeps their drugs in a drawer, by monitoring its closing/opening events it is possible to infer the adherence to medication [22].

As mentioned above, all the sensors used within this research activity were bought in a kit including all of them. This is the usual case when buying commercial devices. The cost of commercial sensor kits may vary in a quite wide range, depending on the type of sensors included in the kit (such as cameras or microphones, or radio-based sensors), the quality of the hardware by means of which sensors have been manufactured and their intrinsic sensitivity, the availability of an app to configure and interact with the sensors, and the possibility to access an online dashboard with reporting functionalities and analytics about the collected sensor data. The usual price may vary from a few tens Euros to several hundred Euros. The specific kit used was bought for around 80 €, a price that can be reasonably considered acceptable in most European countries: it included a PIR sensor, a H&T sensor, three MAG sensors, and a gateway, and a free-to-download Android mobile app to set up and configure the sensors and to visualized the list of collected data. No additional services (such as automatic alerts or reporting) were available for the specific kit chosen. Such a choice was motivated by the aim of showing how even simple and not expensive sensors may give enough information to set up a basic lifelogging solution. This way, within this manuscript, we refer to cheap sensor-based solutions which integrate a few ambient sensors and a basic software application useful to set up the hardware and establish the data transmission to a remote server.

## 3. Technology Acceptance of Sensor-Based Lifelogging Technologies

This section presents previous research on technology acceptance in general and on the acceptance of assisting lifelogging technologies in specific, followed by a short description of the aim and underlying research questions of the present study. 

### 3.1. Technology Acceptance: Concepts & Approaches

The field of technology acceptance research is very broad and not consistent in itself, as it extends across a wide variety of research directions and disciplines. Acceptance as a phenomenon can refer to individual user acceptance towards smaller artifacts (e.g., devices or products) as well as to societal acceptance regarding significant developments (e.g., large-scale technologies) [23]. Since technology acceptance can be viewed from different perspectives, there is no single universally accepted definition of acceptance. In social sciences, however, there is an agreement that acceptance of technological developments is understood not only as the absence of possible resistance but also as active participation and willingness to act or to use [23,24]. There is often a discrepancy between the expectations of users and the functions and services provided by developed technologies and systems. Despite increasing innovative possibilities in the field of technology-based health services, end-user acceptance is often the main obstacle to sustainable implementation and daily use. Research found that the acceptance process is not just a matter of performance, but a complex issue influenced by many user-related, system-related, and context-related factors (e.g., [25,26]). These factors need to be investigated, because—although lifelogging technologies are already used in various contexts—the factors that influence user acceptance are still poorly understood [27].

In order to investigate technology acceptance, diverse acceptance models (e.g., TAM [28] or UTAUT [29]) were conventionally used and adapted. However, these models were originally developed for the investigation of information and communication technologies at the workplace and they do not consider specific characteristics of the respective innovative technology or system. With regard to assisting systems and lifelogging technologies it has proven to be useful to identify specific factors (e.g., usage motives, usage barriers, usage conditions) (e.g., [30]) based on qualitative approaches in a first step. Only then it is useful to quantify these specific factors in subsequent quantitative empirical approaches. 

### 3.2. Acceptance of Lifelogging Technologies

Previous research with regard to the acceptance of assisting technology has focused on the individual, specific technologies in numerous studies. Diverse research results show that the use of assisting technology in old age is predominantly positively evaluated and acknowledged as assistance in everyday life by different user groups [31,32,33]. However, these findings contrast with the fact that actual and sustainable everyday usage of assisting technology in the home environment is still rare [34]. Hence, it is important to investigate potential usage motives and barriers with regard to the use of assisting lifelogging technologies in detail. In this regard, Peek et al. (2014) conducted a meta-analysis of sixteen studies identifying six different thematic areas of acceptance factors, focusing in particular on (expected) benefits and concerns of using assisting technology.

Starting with the motives of using assisting technology, increased safety (e.g., in terms of faster responses in emergencies) was revealed to be a central factor and usage motive in several studies (e.g., [33,35]. In addition to the perceived usefulness of the technology (e.g., [2,36], two other benefits have been identified as important factors influencing the adoption: First, some studies have highlighted increased independence and autonomy as a key motive to use assisting lifelogging technologies [36,37], while second, the relief of family caregivers has been identified as a usage motive (e.g., [33]). With regard to barriers to the use of assisting technology, the analysis by Peek et al. (2014) confirmed that privacy concerns were relevant, in addition to expected high costs (e.g., [35,38]). In more detail, the use of assisting technology in the own home was perceived as an invasion of privacy, and concerns about the potential misuse of data and unauthorized access to data were identified as important barriers. In addition to this very central issue, on the one hand, technology-related factors, such as the feeling of surveillance, feared dependence on technology as well as a lack of trust in technology, were confirmed as possible barriers to the acceptance of assisting technology (e.g., [30]. On the other hand, emotional factors, in terms of feared loneliness or isolation and replacement of human attention by technology, have also been identified as factors influencing acceptance (e.g., [33]).

In addition to fundamental usage motives and barriers, technology-related factors can also influence the acceptance of assisting lifelogging technology. Peek et al. (2014) defined possible alternatives to technology as a factor that negatively influences acceptance. On the one hand, this refers to the possible support in everyday life by family members or caregivers (e.g., [39]). On the other hand, existing and already used technologies (e.g., a home emergency call system) proved to be an obstacle to the use of more complex assistance systems (e.g., [40]). In detail, it must be noted that most acceptance studies examine very specific systems individually. However, there are also studies that have directly compared the acceptance of different assistance systems. In terms of a rough technology classification [41], previous research results show that visually-based technologies are less preferred and accepted than position-determining and audio-based technologies. In line with this, a further study [42] showed that using cameras and microphones is clearly rejected—due to their greater intrusion into the privacy of patients and staff. In comparison, technologies with less intensive data handling (e.g., emergency call switches, fall sensors, room sensors) were significantly more accepted. Confirming these results for a heterogeneous sample, another study [43] found that when given a free choice of technologies, the participants clearly rejected camera-based technologies and rather preferred using already known or even simpler technologies (e.g., emergency call switches, smart watches, smartphones).

Summarizing previous research on the acceptance of lifelogging technologies, the majority of studies focused on video- and audio-based technologies, while only a few studies focused on sensor-based technologies (e.g., ultrasonic whistles [44]). In comparison, there is hardly any specific knowledge about the future users’ perceptions and evaluations of sensor-based lifelogging technologies.

### 3.3. Research Aim and Questions

As presented above, previous research on the acceptance of assisting technologies predominantly focused on video- and audio-based technologies and systems. However, there were comparably few studies investigating specific sensor-based technologies. Therefore, this study aimed at a comparative investigation of three specific sensor-based lifelogging technologies: PIR, H&T, and MAG. The underlying research questions were the following: RQ1: Does the perception of benefits depend on the specific sensor type?RQ2: Does the perception of barriers depend on the specific sensor type?RQ3: Does the acceptance differ for the specific sensor types?

Further, the empirical study is based on a close interdisciplinary collaboration between (biomedical) electric engineering and communication science experts. From a technical perspective, the costs of acquiring sensor-based technologies in a home environment can be realistically quantified. However, it is not known so far, how (high) future users estimate these costs to be and to what extent future users are willing to pay for sensor-based technologies in their home environments. Therefore, the underlying research questions were the following:RQ4: How (high) are the costs for the acquisition of sensor-based technologies estimated to be?RQ5: Does the willingness to pay for the acquisition of sensor-based technologies differ from the assumed costs?

## 4. Methodological Approach

In the following, the methodological approach of the empirical study is presented, introducing the design of the online survey, the applied data analysis procedures as well as the characteristics of the participants.

### 4.1. Design of Online Survey

Figure 1 presents a schematic overview of the online survey’s structure. In the first part, the participants indicated demographic characteristics, such as their age, gender, and highest educational level. Then, the participants answered questions regarding their living situation, before specific health conditions and experiences in care were focused. In more detail, the participants indicated if they suffer from a chronic disease (yes/no answer options) and if they need assistance and support in their everyday life (answer options: never, rarely, regularly, always). Focusing on previous experiences in care (yes/no answer options), the participants were asked if they have professional experience in care, if they have passive private experience (i.e., a close family member is/was in need of care) or if they have active private experience in care (i.e., they care/cared for a close family member themselves). In the second part of the survey, the participants firstly received a short scenario. Within the scenario, all participants were asked to empathize with a future situation imaging that they are in older age, depend on assistance and care in their everyday life, and live alone in their home environment. This way, all participants had the same baseline conditions prior to the evaluation of the sensor types. In a second step, the participants received detailed information about the three sensor-based lifelogging technologies in a randomized order. Thereby, technically correct information was presented as comprehensible and simply as possible using both descriptive text and illustrations in each case based on a close interdisciplinary collaboration between engineering and communication science experts. To ensure clarity, comprehensibility, and unambiguity the whole survey—but in particular the information texts and illustrations—were tested in several pretests prior to the start of the study. Subsequent to receiving the information to one of the three specific sensor-based technologies, the participants were asked to evaluate each time perceived benefits using seven items (PIR: α=0.881; H&T: α=0.884; MAG: α=0.850), perceived barriers using nine items (PIR: α=0.896; H&T: α=0.898; MAG: α=0.894), and the acceptance/intention to use the technology in the own home environment using three items (PIR: α=0.887; H&T: α=0.893; MAG: α=0.901). Each participant evaluated all three sensor-based technologies in a randomized order. All items being used for the assessments can be seen in Table 1. In the final part of the survey, the participants were asked to assume the costs of sensor-based technologies in general. Here, the participants should imagine that their own home (house or apartment) consists of five rooms, and that all these rooms should be equipped with room sensors. The participants evaluated the assumed costs based on a slider between 0 and 500. In addition, the participants were then asked to indicate their willingness to pay for the acquisition of these room sensors. Here, the same slider was used for the participants’ evaluation (range between 0 and 500). These two assessments enable a comparison with the real prices of sensor-based lifelogging technologies. Finally, the participants were able to leave comments and feedback regarding the online survey and its content on a voluntary basis.

### 4.2. Data Analysis

All items referring to perceived benefits, perceived barriers, and the intention to use sensor-based lifelogging technologies were measured on six-point Likert scales (min=1; max=6), and the value of 3.5 represented the mid-point of the scale. Hence, values <3.5 indicated rejection, while values >3.5 indicated acceptance of an item. Reliability analyses (Cronbach’s α>0.7) ensured the quality of all measured constructs (here: perceived benefits, perceived barriers, acceptance). Besides descriptive statistics (means (M), standard deviations (SD), and relative frequencies), repeated measure ANOVAs were used to investigate influences of the three sensor types (PIR, H&T, MAG) on the perceptions and acceptance of using ambient sensor-based lifelogging technologies. Thereby, the F-ratio is reported as a calculated test statistic. Beyond that, correlation analyses were conducted to investigate influences of user factors such as age (Pearson correlations: r) and care experience (Spearman correlations: ρ). Within the illustrated table and figures, a single asterisk (*) indicated a significance level of p<0.05, while a double asterisk (**) indicated a significance level of p<0.01; values above the significance level of p>0.05 are interpreted as not significant (n.s.).

### 4.3. Participants

Overall, N=312 individuals have participated in the study and have filled out the online survey nearly completely. On average, the participants were rather young (M = 32.9; SD = 18.7; Min = 18; Max = 91; median = 23) and the sample contained a higher proportion of females (62.8%; *n* = 196). compared to males (36.9%; n=115), while one participant indicated a diverse gender (0.3%). The sample was educated comparably high with 55.8% (n=174) of the participants holding a university entrance qualification and 28.2% (n=88) holding a university degree. Only small parts of the participants reported lower educational levels, such as secondary or elementary school certificates (16.0%; n=50). Asked for their living situation, most of the participants indicated to live in a partnership or be married (56.4%; n=176), while a third (33.3%; n=104) reported to be single, and small parts to be divorced (1.9%; n=6) or widowed (4.2%; n=13). 4.2% (n=13) did not indicate their living situation. Addressing the context of support in older age and care, the participants were also asked for their health status and previous experiences in care. The majority of the participants indicated to be healthy (79.8%; n=249), while 20.2% (n=63) reported suffering from a chronic disease. Further, the great majority indicated to never depend on support in everyday life (87.5%; n=273), while 10.6% (n=33) need support rarely, and 1.9% (n=6) depend regularly on assistance and support in their everyday life. Asked for experiences in care (yes/no answers), 17.3% (n=54) reported to be professionally experienced in the care, while 24.0% (n=75) have active private experience in caring for a person in need of care, and 31.7% (n=99) have passive private experience in terms of having a family member in need of care. Finally, the participants also indicate previous experiences in using lifelogging technologies and devices (yes/no answers): 64.4% (n=201) reported experiences in health monitoring (e.g., steps, pulse, sleep), 66.7% (n=208) in position tracking via GPS signal, 55.8% (n=174) in life archiving (i.e., collecting pictures, texts, other data), 48.4% (n=151) in the tracking of consumption behavior (e.g., loyalty cards such as Payback for evaluating customers’ purchases and preferences), and 9.3%(n=29) in performance measurement at the workplace (i.e., recording the productivity of workers).

## 5. Results

Within this section, the results of our empirical study are presented, starting with the future users‘ evaluation of the three different sensor-based lifelogging technologies in terms of perceived benefits, perceived barriers, and acceptance (Figure 2, Figure 3 and Figure 4, Table 1). Further, the results referring to assumed costs and the users‘ willingness to pay for the sensors are described (Figure 5).

### 5.1. User Evaluation of Different Sensor-Based Technologies

Starting with the evaluation of perceived benefits (RQ1), the participants evaluated all seven items affirmatively, but significantly different depending on the three sensor-based technologies (Figure 2, Table 1). The benefits of *Recognizing deviations from normal behavior (e.g., leaving but not returning; wandering at night)* (F(2,308)=7.98;p<0.01) and *Recognition of changes in movement (e.g., due to medication intake)*(F(2,306)=9.39;p<0.01) were both evaluated significantly higher for the PIR and MAG sensors compared to the H&T sensors. The benefit of *Triggering alarms (e.g., when objects are not closed properly, in emergency situations)* (F(2,307)=7.49;p<0.01) was evaluated highest for the MAG, followed by the H&T and PIR sensors. Using sensor-based lifelogging technologies for *Automatic reminders (e.g., to ventilate, close doors)* (F(2,308)=17.98;p<0.01) was evaluated more positively for the MAG and H&T sensors compared to the PIR sensors. The most striking differences were found for the evaluation of *Increased security (e.g., forgotten pot on the stove)* (F(2,306)=32.17;p<0.01), which was highest for the H&T sensors, followed by the PIR and MAG sensors. Finally, the benefits of *Recognition of emergencies (e.g., falls)* (F(2,307)=10.21;p<0.01) and *Recognition of emergency situations (e.g., being not able to stand up)* (F(2,306)=11.50;p<0.01) were both evaluated significantly higher for the PIR sensors compared to the MAG and H&T sensors.

Considering individual characteristics of the participants, correlation analyses did not reveal significant relationships with regard to the age of the participants (PIR: r=0.064, p=0.263, n.s.; H&T: r=0.089; p=0.119, MAG: r=0.071, p=0.213, n.s.). Instead, the experience in caring for a family member was slightly related with the perception of benefits (PIR: ρ=−0.131; p=0.021, H&T: ρ=−0.124; p=0.028, MAG: ρ=−0.179; p=0.001), indicating higher evaluations by care-experienced participants.

Moving to the evaluation of potential barriers of using sensor-based lifelogging technologies (RQ2), most of the potential barriers were not evaluated differently depending on the three sensor types (Figure 3, Table 1). Further, the evaluations oscillated around the middle of the scale without any strong agreements or rejections.

The barriers *Perceived violation of own privacy* (F(2,310)=6.031;p<0.01) and *Feeling of surveillance* (F(2,310)=6.36;p<0.01) received both comparatively high agreements and were both evaluated significantly higher for the PIR sensors, followed by the MAG and H&T sensors. Compared to these slightly confirmed aspects, the four barriers *Feeling of dependence* (F(2,310)=0.738;p=0.48, n.s.), *Need for a permanent receiver (=collector)* (F(2,310)=2.46;p=0.09, n.s.), *Doubts about reliability* (F(2,310)=1.16;p=0.32, n.s.), and *Unintentional damage/defects of the sensor (e.g., by dropping)* (F(2,310)=0.04;p=0.96;n.s.) were assessed rather neutrally with values around the middle of the scale (=3.5) independently from the three sensor types. The two statements *Perceived harmfulness of signals to health”*(F(2,310)=2.11;p=0.12, n.s.) and *Fears of emissions (e.g., noise, light, radio)* (F(2,310)=1.64;p=0.19, n.s.) were both slightly rejected to be barriers of using the sensor-based technologies, again independent from the three sensor types. Finally and independent from the three sensor types, the barrier *Unintentionally triggered alarms* (F(2,310)=1.12;p=0.33, n.s.) received the comparatively highest affirmative evaluations.

Considering individual characteristics of the participants also for the perception of barriers, correlation analyses did neither reveal significant relationships with regard to the age of the participants (PIR: r=0.023, p=0.690, n.s.; H&T: r=0.082, p=0.149, n.s.; MAG: r=−0.007; p=0.907, n.s.) nor with regard to the participants’ experience in caring for a family member (PIR: ρ=−0.010, p=0.866, n.s.; H&T: ρ=0.009, p=0.877, n.s.; MAG: ρ=0.028; p=0.628, n.s.).

All three items referring to the acceptance of using sensor-based technologies (RQ3) were evaluated significantly depending on the three sensor types (Figure 4, Table 1). Thereby, the perceived usefulness, acceptance, and willingness to use the MAG and H&T sensors were in tendency higher compared to the PIR sensors: *I find the use of such sensors useful* (F(2,307)=3.50;p<0.05) and *I can well imagine the use of the sensors* (F(2,307)=8.43;p<0.01). Compared to these two items, the most concrete item *I would like to use such sensors in my home* (F(2,307)=7.25;p<0.01) was evaluated less affirmatively, but still higher for the MAG and H&T sensors than the PIR sensors.

Considering individual characteristics of the participants also for the intention to use sensor-based lifelogging technologies, correlation analyses did reveal only partial and slight significant relationships. With regard to age, the results partly indicate that older participants tend to show a slightly higher intention to use the PIR and the MAG sensors compared to younger participants (PIR: r=0.115, p=0.042; H&T: r=0.072, p=0.203, n.s.; MAG: r=0.160; p=0.005). Referring to the participants’ experience in caring for a family member, only the usage intention for the MAG sensor slightly correlated with the caring experience of the participants (PIR: ρ=−0.106, p=0.060, n.s.; H&T: ρ=−0.083, p=0.143, n.s.; MAG: ρ=−0.154; p=0.007).

**Figure 4 sensors-21-08297-f004:**
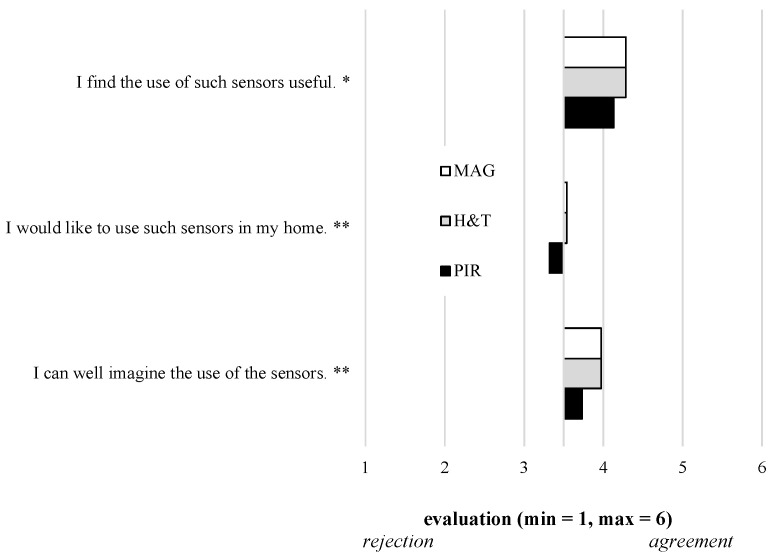
Users’ acceptance of using three different sensor-based lifelogging technologies (RQ3).

**Figure 5 sensors-21-08297-f005:**
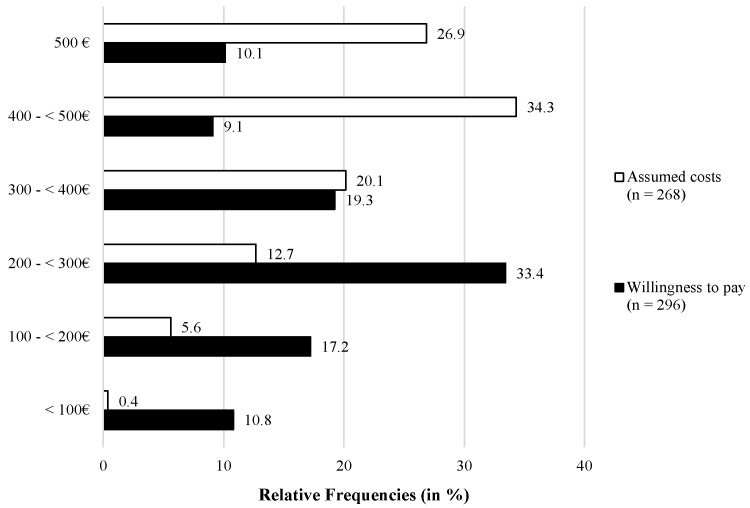
Assumed costs of and willingness to pay for sensor-based lifelogging technologies (RQ4 & RQ5).

### 5.2. Assumed Costs and Willingness to Pay

Besides the evaluation of the acceptance and perception of sensor-based technologies, the participants were asked to assume the actual costs of the described technologies (RQ4). For this purpose, the participants should imagine that they live in an apartment/house with 5 rooms and that they want to equip this apartment/house with room sensors. Following the assumed costs, the participants were also asked to indicate the amount of money they would be willing to pay for the purchase of a sensor-based lifelogging system in their own home environment. Figure 5 shows the results in terms of relative frequencies of the assumed costs and willingness to pay, which the participants were able to indicate specifically within the range of 0–500 €. Overall, the assumed costs amounted to 390.04 € (SD = 106.0; n=268). The majority of the participants assumed the costs to be between 400–500 € (34.3%; n=92) or exactly 500 € (26.9%; n=72), representing the maximum of the range. Further, 20.1% (n=54) selected amounts between 200–300 €, 12.7% (n=34) between 200–300 €, 5.6% (n=15) between 100–200 €, and only 0.4% (n=1) chose amounts lower than 100 €. Compared to that, the average willingness to pay for the sensor-based technologies (RQ5) was clearly lower (but still relatively high) and amounted to 263.88 € (SD = 128.8; n=296). Here, the majority of the participants (33.4%; n=99) were willing to pay between 200–300 €, followed by 300–400 € (19.3%; n=57) and 100–200 € (17.2%; n=51). Lower proportions of the participants indicated to be willing to pay 500 € (10.1%; n=30), between 400–500 € (9.1%; n=27), or less than 100 € (10.8%; n=32).

## 6. Discussion

This study provided first insights into the future users’ perceptions, acceptance, and willingness to pay for specific sensor-based technologies. Thereby, a comparative investigation was enabled based on an interlocking of a technical and a user-centered perspective. Based on the respondents’ assessments of the perceived benefits and barriers for the provided sensors, the potential users generally confirmed the advantages of the sensors, although there are significant differences in places between the sensors studied. Increased security was appreciated especially when using the H&T sensors, and PIR sensors were valued predominantly for the recognition of emergency situations. As to the perceived barriers, respondents primarily apprehended unintentionally triggered alarms and the invasion of privacy, which were especially true for the PIR sensors. They also perceived it as a disadvantage to feel surveilled in their home environments and even cast doubt on the reliability of the sensor-based technologies. Regarding the estimations of the costs and the willingness to pay expenses for the sensors, it appears that the majority of the participants overestimate the costs for this technology. On the other side, many respondents are willing to spend a realistic amount of money for reasonable equipment of sensor-based technology in their homes.

In the following, the results are discussed with regard to previous research in the field differentiating between technical opportunities and functions on the one hand, and the user’s perception, acceptance, and willingness to pay for such developments on the other hand. The discussion on these points is guided by the previously defined research questions. Further, the limitations of this approach are pointed out and the opportunities for necessary and proposed future work are identified.

### 6.1. Comparing User-Relevant and Technical Perspectives

Answering the first research question from the communication science perspective, the results showed that the perception of benefits (RQ1) significantly varied depending on the type of sensor. Here, all single benefit items differed significantly depending on the sensor types. In this regard, previous research (e.g., [41,42]) has so far been limited to findings that acceptance is higher for less privacy-intrusive technologies using binary data (e.g., emergency buttons, room sensors, compared to video- or audio-based technologies). Hence, the gained insights of this study revealed specific differences in the benefit perception of specific sensor-based technologies. To mention some detailed results, the H&T sensor received the highest agreements with regard to the benefit of increased security. These results indicate that the participants understand and acknowledge the technical functionality of this specific sensor and attach higher importance to it compared to both other sensor types. Further, the benefit of recognizing emergencies and respective situations was evaluated highest depending on the PIR sensor, indicating that the participants expect a greater effect from the PIR sensor compared to the MAG or H&T sensor in this regard. It is interesting to consider the results presented under the communication science perspective, from a technical perspective as well. H&T sensors by their same nature are able to detect changes in room temperature and relative humidity: as such, the perception of increased security associated with these sensors appears quite natural, thinking about the risks associated with the use of stove and oven. On the other hand, it is interesting to see how the PIR sensors are associated with the idea of a better capability to detect emergency situations. From a technical perspective, PIR sensors may detect presence and motion in indoor environments, but the capability of detecting a possible emergency condition requires complex processing algorithms that are not usually available by default when buying the sensors. In fact, due to the low information content associated with binary sensors (that are able to distinguish among two states, namely motion/no motion, or presence/no presence), powerful and complex algorithms are needed to extract high-level classification of possible emergencies. In this case, the perceived benefits of PIR sensors are higher than their actual technical capabilities.

Research Question 2 related to the perception of barriers cannot be answered as clearly as RQ1. Most of the items (7/9) were not evaluated to be significantly different by the participants. Merely the two barriers related to privacy concerns and a feeling of surveillance showed differences depending on the sensors: as opposed to this, the barriers were higher confirmed for the PIR and partly also for the MAG sensor, whereas they were less confirmed for the H&T sensor. Here, we assume that the participants refer to a misleading mental model that (passive) infrared sensors operate similarly to the infrared cameras and thus they apprehend a potentially higher privacy violation. From a technical perspective, data generated from PIR, MAG, and H&T sensors can expose private information only in the case they are semantically enriched by additional information, such as the specific position in which each sensor is located inside the living environment. In fact, variations in H&T data detected from a sensor located in the kitchen have a different meaning than variations detected from the same sensor located in the bathroom or in a bedroom. The same applies to PIR and MAG sensors as well: each of them gains a potential capability to disclose private information when coupled with a contextual label. In this case, users’ perception about the risk of privacy disclosure associated with PIR sensors is higher than their intrinsic technical capability.

With regard to the acceptance of sensor-based technologies (RQ3), the results showed slight but significant differences depending on the sensors: thereby, the intention to use was higher for the H&T and the MAG sensors compared to the PIR sensor. Here, it can be assumed that, on the one hand, the MAG and H&T sensors were perceived to be less privacy-intrusive than the PIR sensor. On the other hand, the participants expected the highest increased security when the H&T sensor is used. Both aspects could have led to the slightly higher preferences for the H&T and the MAG sensor compared to the PIR sensor. From a technical point of view, motivation for the fact that H&T and MAG sensors were associated with higher perceived usefulness, acceptance, and willingness to use, with respect to PIR sensors, could be twofold. On the one hand, data generated by H&T and MAG sensors have a direct and clear correspondence with common events or conditions happening or performed within the home environment (e.g., a stove switched on/off, or a door opened/closed), whereas the PIR sensor activation is generically due to motion inside the environment. This could reduce the perceived impact of the PIR sensor data on capturing ongoing events. On the other hand, as already discussed above, PIR sensors are deemed less privacy-preserving sensors than MAG and H&T ones, and this may affect their acceptance and willingness to use them.

In the next step, the differences between the costs assumed by the participants (RQ4) and the real costs to integrate sensor-based technologies in home environments are discussed. Before estimating the costs, the participants should imagine that they would like to equip their own home—for reasons of better comparability consisting of five rooms— with sensor-based technologies. Overall, the participants assumed the costs to be comparably high (almost 400 €), higher than the cost needed to cover five rooms with the sensors belonging to the same kit tested. In fact, assuming to install a PIR, MAG, and H&T sensor in each room, five sensors of each type are needed and a single gateway, for a total cost of around 250 €. From a technical perspective, the sensor kit chosen for our study allows for self-installation, which is a quite common feature of most of the solutions available in the market. In fact, most of the manufacturers aim to sell *ecosystems* of sensors that can be easily put in place by the customers, without requiring a technical intervention (which is different from the case of installing a professional surveillance and security system): starting from a basic configuration, usually including a minimum number of sensors and a gateway, the user has the possibility to add new compatible sensors in the future. Buying sensors and devices from the same manufacturer ensures seamless integration of the new components into the existing solution already in place. Thanks to this approach, basically motivated by marketing strategies, a user may first go for a basic sensor set in one or two rooms, and then decide to enrich the system by covering additional rooms with new devices. By adding new devices onto the existing basic installation, the cost increases only marginally, because the same gateway may serve a huge number of sensors. The installation of additional gateways may be necessary only in the case of quite large environments, to ensure radio coverage for the BLE-enabled sensors.

As the last aspect, the participants’ willingness to pay for sensor-based technologies (RQ5) is discussed in relation to the real costs. Not surprisingly, the participants’ willingness to pay was clearly lower than the assumed costs. However, reaching on average 260 €, the willingness to pay for the integration of sensor-based technologies in the own home environment was still comparably high and, interestingly—from a technical perspective —enough to cover the true cost associated with the installation of a basic sensor set in five rooms (see Section 2.1).

### 6.2. Limitations and Future Work

Beyond the gained insights into the future users’ acceptance of specific sensor-based technologies, some limitations with regard to the methodological approach and sample of our empirical study have to be considered for future research in the field.

Starting with methodological issues, it has to be mentioned that we analyzed the whole sample of participants so far. Further, we considered first correlation analyses with regard to some user factors (e.g., age, care experience) indicating only single and slight relationships. As there is a high heterogeneity among the users of sensor-based technologies, our future research will investigate the potential influence of user diversity and other individual characteristics (e.g., gender, health status, technical expertise) on the acceptance and perceptions of sensor-based technologies.

Further, it has to be mentioned that a scenario-based approach was applied to evaluate the acceptance of sensor-based lifelogging technologies by means of an online survey. Future studies should focus on assessments and evaluations of users’ real experience with using the different sensor types for a certain time. Then, it would be possible and interesting to compare the survey-based acceptance with the real hands-on experience results.

In regard to this, future investigations could benefit from the design of a sample, low-cost sensor system for data collection, based on free development platforms such as the ESP8266-IoT, which could be used to assess users’ real experience on the same technical set up.

The next aspect refers to the information provided to the participants about the three sensor-based technologies. Of course, provided information has always the potential to shape the participants’ evaluations. To control the given information, we checked the technical correctness, clarity, comprehensibility, and ambiguity of all descriptions and explanations based on the interdisciplinary collaboration and by conducting pretests prior to the start of the study.

Further, also the sample of our study has a few peculiarities that need to be mentioned. The sample was on average rather young, contained a higher proportion of female than male participants, and was comparably highly educated. The representativeness, which is a prerequisite to being able to infer a population, is thus not fully met in the present study. Hence, future studies should try to reach more balanced samples with regard to age, gender, and educational level. In order to address in particular older and frail people, who could profit most from using sensor-based technologies in terms of support and security, it would be useful to focus on older participants, individuals with a poor health condition, and/or persons in need of support and care.

Finally, it has to be mentioned that the participants came from one single country—Germany. Future studies should aim at reaching samples from different countries in order to compare cultural similarities and differences in the perception of sensor-based technologies.

## Figures and Tables

**Figure 1 sensors-21-08297-f001:**
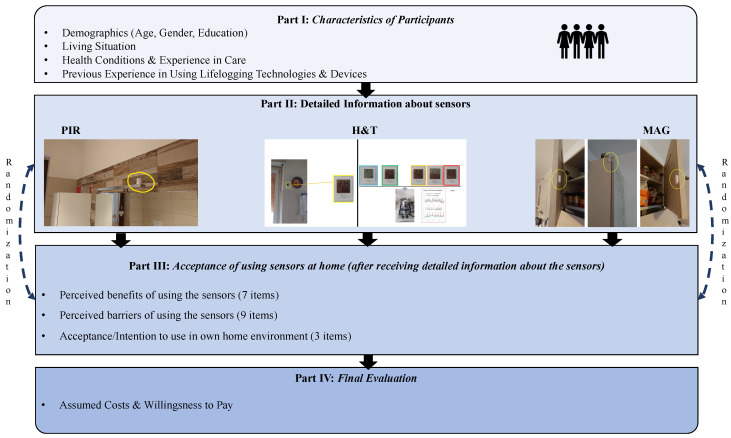
Structure of the online survey.

**Figure 2 sensors-21-08297-f002:**
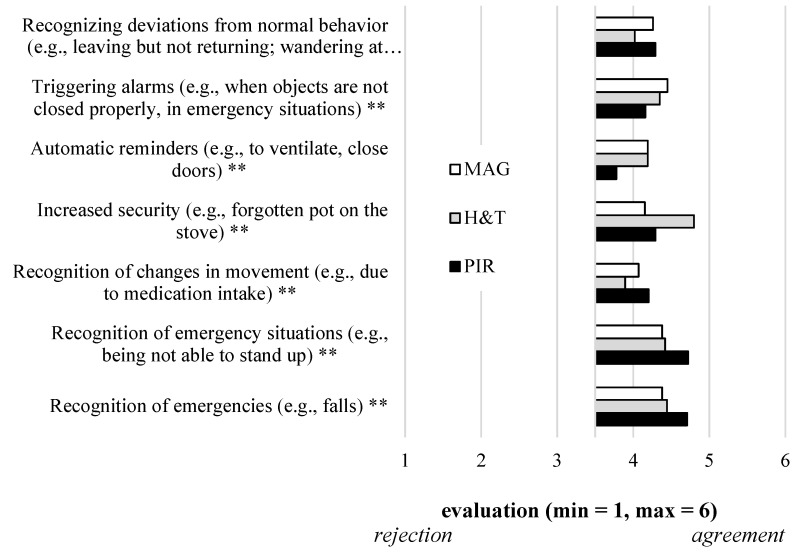
Users’ evaluation of perceived benefits of using three different sensor-based lifelogging technologies (RQ1).

**Figure 3 sensors-21-08297-f003:**
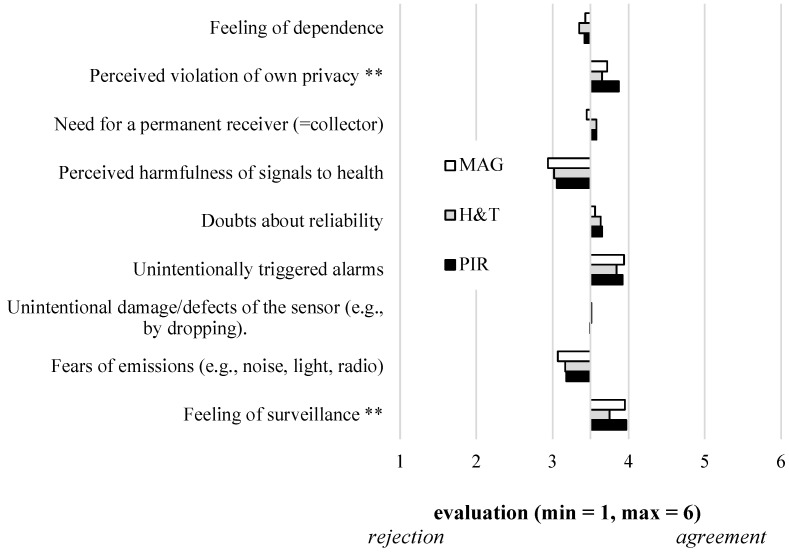
Users’ evaluation of perceived barriers of using three different sensor-based lifelogging technologies (RQ2).

**Table 1 sensors-21-08297-t001:** Descriptive statistics of the participants‘ evaluation of the three sensor-based lifelogging technologies.

Construct	Operationalization	Sensor-Based Technology							
		Overall		PIR		H&T		MAG	
		Min	Max	M	SD	M	SD	M	SD
**Perception of** **Benefits**	Recognizing deviations from normal behavior (e.g., leaving but not returning; wandering at night) **	1	6	4.3	1.3	4.0	1.4	4.3	1.4
	Triggering alarms (e.g., when objects are not closed properly, in emergency situations) **	1	6	4.2	1.4	4.4	1.4	4.5	1.3
	Automatic reminders (e.g., to ventilate, close doors) **	1	6	3.8	1.5	4.2	1.4	4.2	1.4
	Increased security (e.g., forgotten pot on the stove) **	1	6	4.3	1.4	4.8	1.2	4.2	1.5
	Recognition of changes in movement (e.g., due to medication intake) **	1	6	4.2	1.2	3.9	1.4	4.1	1.3
	Recognition of emergency situations (e.g., being not able to stand up) **	1	6	4.7	1.1	4.4	1.4	4.4	1.4
	Recognition of emergencies (e.g., falls) **	1	6	4.7	1.2	4.4	1.5	4.4	1.4
**Perception of** **Barriers**	Feeling of dependence	1	6	3.4	1.4	3.4	1.4	3.4	1.4
	Perceived violation of own privacy **	1	6	3.9	1.5	3.7	1.5	3.7	1.6
	Need for a permanent receiver (=collector)	1	6	3.6	1.3	3.6	1.4	3.5	1.4
	Perceived harmfulness of signals to health	1	6	3.1	1.4	3.0	1.4	2.9	1.4
	Doubts about reliability	1	6	3.7	1.3	3.6	1.3	3.6	1.3
	Unintentionally triggered alarms	1	6	3.9	1.3	3.8	1.3	3.9	1.4
	Unintentional damage/defects of the sensor (e.g., by dropping).	1	6	3.5	1.3	3.5	1.3	3.5	1.3
	Fears of emissions (e.g., noise, light, radio)	1	6	3.2	1.4	3.2	1.4	3.1	1.4
	Feeling of surveillance **	1	6	4.0	1.2	3.8	1.5	4.0	1.6
**Acceptance of** **Assistive Technology**	I find the use of such sensors useful. *	1	6	4.2	1.2	4.3	1.2	4.3	1.2
	I would like to use such sensors in my home. **	1	6	3.3	1.4	3.5	1.4	3.5	1.4
	I can well imagine the use of the sensors. **	1	6	3.7	1.4	4.0	1.4	4.0	1.4

## Data Availability

The data presented in this study are available on request from the corresponding author. The data are not publicly available due to privacy restrictions.

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
