# Peer review of "Acceptance and Preferences of Using Ambient Sensor-Based Lifelogging Technologies in Home Environments"

_sensors, 2021, doi:10.3390/s21248297_

Round 1

Reviewer 1 Report

  • very interesting topic and approach taken.
  • while chapter 1 and 2.1 remain focused on announced sensor-based (PIR, H&T, MAG) technologies, chapter 2.2.2 goes into more broader discussion almost typical for introduction chapters
  • chapter 2 mixes a lot of different subchapters to authors may possibly consider the alternative structure of the paper
  • by analyzing all different and highly heterogeneous user groups all at once the overall results and conclusions get blurred. For example do all respondents consider themselves as residents in the smart home or younger respondents consider themselves as caregivers? Different age groups have different use cases and expectations so it may be very interesting to see how different age groups responded and how their responses compare (and change). Authors also recognize this and put it in future work the question remains are the results as currently described valuable for anything and for whom exactly?
  • also do the sensors mentioned imply also the systems that use them (the price of sensors is only a fraction of the total cost of a system that integrates them and that actually processes and analyzes the obtained data to produce the desired outcome such as behavior patterns and alarms..). Futhermore the same price can in different countries (with different average incomes) be considered different (acceptable in one country can mean too expensive in another)
  • Since the results are from Germany only suggestion is to put this info  immediately in the abstract, not to wait the end of the conclusion
  • unclear why "Feeling of independence" is stated as barrier.
  • could be elaborated in more detail:
    • what ** mean in table
    • what F() represent
    • rejection vs acceptance
  • text explanations should follow the order of items as depicted in figures for easier reading

Author Response

Response to Reviewer 1 Comments

Point 1: Very interesting topic and approach taken.

Response 1: Thank you very much for your feedback.

Point 2: While chapter 1 and 2.1 remain focused on announced sensor-based (PIR, H&T, MAG) technologies, chapter 2.2.2 goes into more broader discussion almost typical for introduction chapters.

Response 2: Thank you very much for your feedback. For a better overview, we highlighted undertaken changes in blue within the revised version of our manuscript.

Point 3: Chapter 2 mixes a lot of different subchapters to authors may possibly consider the alternative structure of the paper.

Response 3: Thanks for this suggestion. We changed the structure of the paper by differentiating between the technical (new section 2) and the acceptance-related background (new section 3). Now, the contents are significantly more clearly structured for the reader.

Point 4: By analyzing all different and highly heterogeneous user groups all at once the overall results and conclusions get blurred. For example do all respondents consider themselves as residents in the smart home or younger respondents consider themselves as caregivers? Different age groups have different use cases and expectations so it may be very interesting to see how different age groups responded and how their responses compare (and change). Authors also recognize this and put it in future work the question remains are the results as currently described valuable for anything and for whom exactly?

Response 4: Thank you very much for this valuable comment. First of all, we would like to comment on our methodological procedure in more detail. All participants received a short scenario in which they were asked to imagine themselves to be older, to live alone, and to be dependent on assistance in everyday life. Hence, all participants had to empathize with the same situation in order to keep the external conditions as constant as possible. To make this clearer for the reader, we added a paragraph in the section “Design of the Online Survey'', providing more detailed information on our methodological procedure. The second aspect you mentioned refers to the potential influences of user diversity factors, such as age or care experience: In order to provide first insights into potential effects of factors such as age and care experience, we additionally conducted correlation analyses (related to the examined constructs, such as perceived benefits, perceived barriers, and intention to use the PIR, H&T and MAG sensors) and we added these results as short notices to the respective results sections. However, the results show only isolated and weak correlative relationships in this regard, and no patterns of particularly strong influences of age or other diversity characteristics emerge. Keeping the constant baseline conditions based on the applied scenario as well as the results of the correlation analyses in mind, the results are valuable and valid for the whole sample of participants.

Point 5: Also, do the sensors mentioned imply also the systems that use them (the price of sensors is only a fraction of the total cost of a system that integrates them and that actually processes and analyzes the obtained data to produce the desired outcome such as behavior patterns and alarms..).

Response 5: We thank the Reviewer for this comment. In our study, we focus on sensor-based solutions that could potentially be installed by skilled users on their own, or by a technician, but without the need for a strong and specific expertise. This way, we exclude from the study those high-level and professional solutions that are typically used for video-surveillance and security aims. Most of the sensor kits of the former kind are sold in the market as a combination of hardware devices and companion software. The latter is typically provided in the form of a free app for mobile devices, such as smartphones, which is able to connect to a remote server to which sensors deliver their data, and generate simple analytics and statistical figures regarding, for example, the frequency of activation of a specific sensor in a time interval. The app is usually available for free from the most common appstores while, in some cases, the access to remotely stored data may be subjected to a fee, which is usually set on a monthly or yearly basis. But there are also many available solutions in which the app is completely free to use, such as in the case of the sensor kit used in our study, but added value functionalities, such as reporting or analytics, are not provided. This way, it is possible to say that we refer to cheap sensor-based solutions which integrate sensors and a basic software application. We added a sentence to clarify this aspect in the text, at the end of Subsection 2.3.

Point 6: Futhermore, the same price can in different countries (with different average incomes) be considered different (acceptable in one country can mean too expensive in another).

Response 6: We thank the Reviewer for this insightful comment, which we agree on. Of course, differences in incomes, but also in cultural or societal aspects, may impact the way the sensor system may be “perceived”. In this case, as the research project originally focused on users from European countries (the survey of this study addressed “only” people from Germany) the price of the chosen sensor kit (80 €) was considered to be reasonably “low cost” for most of the target countries. A short sentence has been added in the text, in Subsection 2.3 and we clarified in the abstract that the sample involved participants from Germany.

Point 7: Since the results are from Germany only suggestion is to put this info  immediately in the abstract, not to wait the end of the conclusion.

Response 7: Thanks for this advice. We put this information in the abstract as suggested.

Point 8: Unclear why "Feeling of independence" is stated as barrier.

Response 8: Thank you very much for your comment – we inadvertently mixed up the terms here, intending to say “feeling of dependence” at this point. We corrected the terms within the revised version of the manuscript.

Point 9: could be elaborated in more detail: what ** mean in table; what F() represent; rejection vs acceptance.

Response 9: Thank you very much for your comments referring to the representation of statistics. We added a paragraph within the section “Data Analysis”, explaining the meaning of using the asterisks (**), the F-ratio (F()) and the terms “rejection vs. acceptance”.

Point 10: Text explanations should follow the order of items as depicted in figures for easier reading.

Response 10: Thank you for your feedback. Usually, we would agree that text explanations should follow the order of items. Here, based on the complexity of comparing three different sensor types, we decided to report the results based on evaluation patterns for some items. Nevertheless, we adapted the order if possible.

Reviewer 2 Report

The manuscript reviewed presents an acceptance and preferences of using ambient sensor-based life logging technologies in home environments.

In the related work, in the introduction, the spectrum of the technological developments from video-based approaches could be updated with recent reviews dedicate to fall detection using multiple cameras and artificial intelligent. I believe this would further strengthen the related work and lend support to the methodology applied.

This empirical study was presented with a technical overview of sensor-based technologies taking the PIR, temperature and humidity, and magnetic sensors as examples. In the case of the PIR sensor, in line 96-98, it’s recommended to updated recent reviews on data collection, sending data to online platforms and integration with the WiFi module as following document: a low cost presence detection system for smart homes.

The authors may be able to specify the H&T model used in this study, as there are a wide range of temperature and humidity sensors. In this survey, are you mentioned the technical characteristics of the H&T sensor? This could modify the results since the H&T sensors their functionality is better known than the other sensors and received highest agreements with regard to the benefit of increased security.

In this study, a specific kit was used to collect data. The authors look the future possibility of carrying out a new work with a low-cost system for data collection using a free development platform like: esp8266-IoT or another embedded system?

In Table 1, Descriptive statistics of the participants' evaluation of the three sensor-based lifelogging technologies, what does the single and double asterisk (*) refer to?

I agree with the authors to be able to carry out a future survey but with other countries to buy differences and similarities in the perception of sensor-based technologies, from experience, I think they will have very good research results.

This research article on the use of environmental sensors Lifelogging technologies in home environments, it could be very interesting for future work, to implement experiments with people on the site, that is, that people can use these sensors for a certain time and evaluate their experience.

Author Response

Response to Reviewer 2 Comments

Point 1: The manuscript reviewed presents an acceptance and preferences of using ambient sensor-based life logging technologies in home environments.

Response 1: Thank you very much for your feedback. For a better overview, we highlighted undertaken changes in blue within the revised version of our manuscript.

Point 2: In the related work, in the introduction, the spectrum of the technological developments from video-based approaches could be updated with recent reviews dedicate to fall detection using multiple cameras and artificial intelligent. I believe this would further strengthen the related work and lend support to the methodology applied.

Response 2: We thank the Reviewer for this suggestion. In order to include more recent literature dealing with fall detection using multiple cameras and artificial intelligence, new references (namely 6-8) have been included, and a new sentence added into the text.

Point 3: This empirical study was presented with a technical overview of sensor-based technologies taking the PIR, temperature and humidity, and magnetic sensors as examples. In the case of the PIR sensor, in line 96-98, it’s recommended to updated recent reviews on data collection, sending data to online platforms and integration with the WiFi module as following document: a low cost presence detection system for smart homes.

Response 3: We thank the Reviewer for this comment. While the suggested paper has been added as a new reference in the manuscript, we would like to highlight the fact that our work does not focus on the technical aspects of ambient sensor systems for personal indoor monitoring, but on the acceptance and awareness issues associated with the use of these sensor systems in lifelogging. This is the reason why an extended discussion of technical solutions is not addressed within this manuscript.

Point 4: The authors may be able to specify the H&T model used in this study, as there are a wide range of temperature and humidity sensors. In this survey, are you mentioned the technical characteristics of the H&T sensor? This could modify the results since the H&T sensors their functionality is better known than the other sensors and received highest agreements with regard to the benefit of increased security.

Response 4: We thank the Reviewer for this comment. In our investigation, specific technical details about the H&T sensor used were not provided to survey respondents, but pictures of the sensor were shown to them and the general functionalties were described, as explained in Figure 1 of the manuscript. As already stated in reply to the previous comment, the focus of our study was not really on the technical details of the sensor deployment, but on investigating users’ perception under different perspectives. Basically, any T&H sensor could fit the scope of our analysis.

Point 5: In this study, a specific kit was used to collect data. The authors look the future possibility of carrying out a new work with a low-cost system for data collection using a free development platform like: esp8266-IoT or another embedded system?

Response 5: We thank the Reviewer for this comment and the interesting idea about the possibility to deploy a low-cost system for data collection using free development platforms, like the one mentioned. As already stated in reply to previous comments, this was not the true focus of this study, but of course it could be included among the possible future developments, as a way to set up a kind of sample system to even test the potential users’ perceptions. A sentence has been added in this regard in Subsection 6.2.

Point 6: In Table 1, Descriptive statistics of the participants' evaluation of the three sensor-based lifelogging technologies, what does the single and double asterisk (*) refer to?

Response 6: Thank you very much for your comment. We added an explanation within the section “Data Analysis”, explaining the meaning of the single and double asterisk.

Point 7: I agree with the authors to be able to carry out a future survey but with other countries to buy differences and similarities in the perception of sensor-based technologies, from experience, I think they will have very good research results.

Response 7: Thank you for your comment. We totally agree with you which is why we highlighted this aspect at the end of the discussion section.

Point 8: This research article on the use of environmental sensors Lifelogging technologies in home environments, it could be very interesting for future work, to implement experiments with people on the site, that is, that people can use these sensors for a certain time and evaluate their experience.

Response 8: Thank you very much for your feedback and this suggestion. We integrated it as a possible idea of future research within the “Limitations and Future Research” section.
